# Metabolomics: A Tool to Understand the Impact of Genetic Mutations in Amyotrophic Lateral Sclerosis

**DOI:** 10.3390/genes11050537

**Published:** 2020-05-11

**Authors:** Débora Lanznaster, Charlotte Veyrat-Durebex, Patrick Vourc’h, Christian R. Andres, Hélène Blasco, Philippe Corcia

**Affiliations:** 1UMR 1253, iBrain, University of Tours, Inserm, 37000 Tours, France; c.veyratdurebex@chu-tours.fr (C.V.-D.); patrick.vourch@univ-tours.fr (P.V.); christian.andres@univ-tours.fr (C.R.A.); helene.blasco@univ-tours.fr (H.B.); philippe.corcia@univ-tours.fr (P.C.); 2CHU de Tours, Service de Biochimie et Biologie Moléculaire, 37000 Tours, France; 3CHU de Tours, Service de Neurologie, 37000 Tours, France

**Keywords:** ALS, metabolomics, genetics, iPSC, personalized medicine

## Abstract

Metabolomics studies performed in patients with amyotrophic lateral sclerosis (ALS) reveal a set of distinct metabolites that can shed light on the pathological alterations taking place in each individual. Metabolites levels are influenced by disease status, and genetics play an important role both in familial and sporadic ALS cases. Metabolomics analysis helps to unravel the differential impact of the most common ALS-linked genetic mutations (as *C9ORF72*, *SOD1*, *TARDBP,* and *FUS*) in specific signaling pathways. Further, studies performed in genetic models of ALS reinforce the role of TDP-43 pathology in the vast majority of ALS cases. Studies performed in differentiated cells from ALS-iPSC (induced Pluripotent Stem Cells) reveal alterations in the cell metabolism that are also found in ALS models and ultimately in ALS patients. The development of metabolomics approaches in iPSC derived from ALS patients allow addressing and ultimately understanding the pathological mechanisms taking place in any patient. Lately, the creation of a “patient in a dish” will help to identify patients that may benefit from specific treatments and allow the implementation of personalized medicine.

## 1. Introduction

Amyotrophic lateral sclerosis (ALS) is the most common motor neuron disease in adults, with an incidence of around two per 100,000 persons/year [1]. ALS is characterized by the progressive loss of both upper and lower motor neurons, and patients usually die after 3–5 years of symptoms onset. The pathogenesis of ALS is only partially understood, with several studies highlighting that aberrant RNA metabolism and protein aggregation, glutamatergic toxicity, oxidative stress, and deficits in axonal transport can lead to motor neuron death. Among all the hypotheses put forward to explain the disease, the major role of genetics is worldwide accepted with more than 50 genes currently linked to familial forms of ALS (fALS) that account approximately for 10% of ALS cases [2]. Most of ALS causative genes are linked to the pathogenic mechanisms described before [3]. Interestingly, in the other 90% of ALS cases (sporadic, sALS with no familial hereditary component involved), similar pathogenic mechanisms are observed, and a non-negligible proportion of these cases presents deleterious mutations in the same genes [4].

Such pathogenic mechanisms produce metabolic alterations in each affected cell (as the main target in ALS, motor neurons, but also surrounding cells such as astrocytes) that eventually reflect in the whole organism. For example, it is well known that most of the ALS patients present a hypermetabolic state [5,6,7,8], although a recent meta-analysis showed some controversy in this matter [9]. “Omics” approaches are capable of detecting these alterations in the whole organism. In this sense, metabolomics is capable of investigate the whole metabolome, and the metabolomics profile can highlight a particular metabolic pathway that is under or over-regulated in the patient or a group of patients. Moreover, metabolomics analyses reveal pathways that are commonly altered in sporadic patients with no genetic mutation detected and in known genetic-linked ALS cases, highlighting common pathogenic mechanisms. Further, it can help to identify subgroups of ALS patients—with and without mutations—that could benefit from a particular therapy. This knowledge is important to (1) identify biomarkers for ALS; (2) define homogeneous subgroups at the metabolomics level; and (3) identify patients with alterations in metabolism that could be compensated by targeted therapy.

In this review, we will discuss the most common mutations linked to ALS and present the studies that report metabolic alterations associated with these genetic findings in patients and models of ALS. As metabolomics identifies targets that might be attractive for therapeutic manipulation, we will also discuss some studies that evaluated the possible therapeutic effect of targeting and compensating the alterations found in the metabolome. Finally, studies performed in induced Pluripotent Stem Cells (iPSC) derived from ALS patients reproduce the same metabolic alterations found in the patient. The development of iPSC from ALS patients will help to create the “ALS patient in a dish”, to study drug response and to increase our understanding about the pathological mechanisms linked to each case. 

## 2. Overview of ALS Genetics

Almost all familial cases of ALS are inherited in an autosomal dominant manner [2]. Genome-wide association studies (GWAS) and next-generation sequencing techniques allowed the analysis of very large datasets, contributing to the identification of around 50 potentially causative or disease-modifying genes linked to ALS. Among these, four genes are more frequently associated with ALS [10]: these include variants of *C9ORF72*, *SOD1*, *TARDBP,* and *FUS*, which account for approximately 55.5% of fALS and 7.4% of sALS cases in Europe [11]. Low-frequency variants could contribute significantly to these apparent sporadic cases, but low-allele frequencies may not be captured by current genotyping arrays, and effect sizes may not be large enough to detect it by linkage analysis in families [12]. Moreover, relatively rare but potentially deleterious variants can hinder the determination of their pathogenicity [11].

### 2.1. C9ORF72

Hexanucleotide repeat expansions (GGGGCC) in the intron 1 of chromosome 9 open reading frame 72 (*C9ORF72*) gene are the most common inherited cause of ALS in Europe, accounting for 34% of fALS and 5% of sALS (Table 1) [4]. While the normal gene presents 5 to 10 copies of these hexanucleotide repeats, the *C9ORF72* mutation found in ALS patients presents hundreds to thousands of repeats [13,14]. Although the exact role of the C9ORF72 protein is not well understood yet, some reports suggest its involvement in the autophagy–lysosome pathway [15,16,17]. ALS patients present decreased levels of C9ORF72 protein and mRNA, leading to the hypothesis that a loss of function of this protein could be pathological. However, several studies report that the knockdown of *C9ORF72* in mouse and ALS-derived iPSC does not cause cell degeneration [18,19,20,21,22,23]. Two other proposed pathological mechanisms involve the gain of function: first, RNA toxicity induced by sense, and secondly, anti-sense RNA that sequesters RNA-binding proteins and form foci in the cells [18,24,25]. Second, repeat-associated non-ATG translation of dipeptide repeat proteins (DPRs) inclusions cause toxicity to neurons [26,27,28,29,30,31,32].

### 2.2. SOD1

The protein SOD1 (or Cu/Zn-superoxide dismutase 1) plays an important antioxidant role in the cell. Located in the cytoplasm and in the intermembrane space of the mitochondria (mSOD1), this protein neutralizes the superoxide species produced during the cellular respiration [33]. The involvement of *SOD1* mutations in ALS was first described in 1993 [34]. Since then, studies point out that *SOD1* mutations account for 15%–20% of fALS and around 2% of sALS cases [4], but there is controversy regarding if all mutations are indeed pathogenic [35] (Table 1). Around 185 disease-associated variations in the *SOD1* gene have been identified (the majority being missense mutations) [36]. Disease duration, severity, and site of onset (bulbar or limb) differ significantly and depend on the variants involved (variants and their phenotypes were reviewed elsewhere [11]).

### 2.3. TARDBP

Transactive Response DNA-binding protein 43 (TDP-43) is a protein of 43 kDa encoded by the *TARDBP* gene. TDP-43 binds to pyrimidine-rich DNA and RNA motifs and is implicated in multiple steps of transcriptional and posttranscriptional regulation. To date, at least 48 pathogenic variants in *TARDBP* have been associated with ALS [37]. Even if mutations in *TARDBP* account for 3% of fALS and around 1.5% of sALS (Table 1), 97% of all ALS cases present degenerated motor neurons containing cytoplasmic aggregates rich in TDP-43, making these aggregates the key hallmark for ALS. Furthermore, TDP-43 aggregates were described in fALS caused by mutations in other 19 genes (as *SOD1* and *C9ORF72*, for example) [38]. These aggregates consist of aberrantly phosphorylated and ubiquitinated full-length TDP-43, as well as 35- and 25-kDa C-terminal fragments of the protein [39,40]. While much has been discussed whether TDP-43-associated neurotoxicity is due to a potential loss of function or gain of function, more and more research point to the central role of TDP-43 aggregates in mediating the motor neuron death observed in ALS patients [38].

### 2.4. FUS

Fused in sarcoma (FUS) is a ubiquitously expressed RNA binding protein found aggregated in ALS patients with pathogenic variants in the *FUS* gene. The first variants were identified in 2009 and are frequently associated with early onset and juvenile ALS [41,42,43,44,45]. Today, more than 50 autosomal dominant variants were identified (from missense mutations to nonsense mutations) [37]. Interestingly, FUS cytoplasmic aggregates are only found in FUS-ALS patients, and these patients do not present TDP-43 aggregates [38,42]. As TDP-43, FUS is predominantly a nuclear protein that can shuttle between the nucleus and the cytoplasm [46], and many of the pathogenic variants are found within the nuclear localization signal of the FUS protein, leading to the redistribution of FUS to the cytoplasm [47,48] (Table 1).

**Table 1 genes-11-00537-t001:** Metabolomics findings from studies performed in genes associated with amyotrophic lateral sclerosis (ALS) cases.

Gene	Function of Coded Protein	fALS/sALS Cases (%)	Alteration in Metabolome	Model
*C9ORF72*	Autophagy–lysosome pathway	34/5	↓HDL	FTLD [49]
*SOD1*	Antioxidant	15-20/2	↓ aminoacids ↓ aminoacids; ↑ glycolysis ↓ glutamate ↑ putrescine and spermidine; ↓ hydroxyproline ↑ creatinine	ALS patients [50] NSC-34 cells [51] Motor neuron/ astrocytes cultures [52] mice [53] ALS patients [53]
*TARDBP*	RNA metabolism	3/1.5	↓ carnitine and beta-hydroxybutyrate ↑ phosphoenolpyruvate and pyruvate ↑ fatty acids	Drosophila [54] Drosophila [55] HEK293T cells [56]
*FUS*	RNA metabolism	2.4/0.16 [57] 3.8 [58] 4.1 [42]	none	iPSC-derived motor neurons [59]
SNP rs1985243	not described	-	↑ gamma-glutamylphenylalanine	ALS patients [60]

fALS: familial ALS; FTLD: frontotemporal lobar degeneration; sALS: sporadic ALS; SNP: single nucleotide polymorphisms.

## 3. Metabolomics and Genetic-Linked ALS: A Way into the Targeted Treatment

Metabolomics evaluates a vast array of metabolites in a biological sample, thus reflecting the overall metabolome of an individual at a given time. For example, metabolites levels are influenced by food and lifestyle choices, the environment, disease mechanisms, and exposure to drugs or pharmacological treatments [61]. With this in mind, it is logical to hypothesize that the genetic background of an individual may play a role in the metabolic profile. Further, it can help to identify diseases subtypes according to the different genetic background and mutations involved in the pathogenic process (Figure 1). Metabolomics can also be used to follow drug response, as metabolites will reflect how the organism responds to selected drug candidates. Moreover, as metabolomics reflects alterations in the physiology, it can also show which pathways are dysfunctional and need to be compensated, opening new perspectives for metabolomics to be applied with therapeutic purposes. Some clinical trials applied metabolomics with these purposes are also discussed below.

### 3.1. C9ORF72

Despite its wide involvement in fALS, few metabolomics studies have been conducted in *C9ORF72* models. A metabolomics study performed on sera of patients with *C9ORF72* mutation (only patients with frontotemporal lobar degeneration (FTLD), no ALS patients), revealed lower HDL cholesterol in *C9ORF72* patients versus non-mutated patients [49]. The pathogenicity of hexanucleotide repeat expansions in *C9ORF72* has been mainly studied in Drosophila and zebrafish models. However, the emergence of the use of iPSC in ALS research opens the perspective of using cellular models from patients’ cells. Interestingly, it was shown that iPSC-derived motor neurons from C9ORF72-linked ALS/FTD patients present higher levels of free fatty acids and liquid droplets than controls, indicating a dysfunctional lipid metabolism [62]. Although no study so far correlated lipid metabolism and the onset of behavioral and cognitive symptoms in ALS/FTLD patients, it is possible that the alterations in lipid metabolism are associated with the appearance of such symptoms. For example, dysfunction in lipid metabolism was linked to cognitive impairment in several types of dementia and Alzheimer’s disease [63,64,65,66]. Further studies should be conducted in ALS/FTLD patients carrying the *C9ORF72* repeat expansions—even from the asymptomatic phase—to improve our understanding about the pathological mechanisms taking place. 

### 3.2. SOD1

One striking example of metabolome profile linked to a genetic mutation was reported in the study of Wuolikainen and co-workers [50]. They analyzed the metabolomics profile in the cerebrospinal fluid (CSF) of ALS patients and found a different signature of metabolites in ALS patients bearing different *SOD1* mutations (including S105L, D101G, A89V, G93S, I113F, and D90A) in comparison to patients without *SOD1* mutations (both familial and sporadic cases). Further, they described that both homozygous and heterozygous carriers of the D90A mutation presented a very distinct metabolic signature, behaving as a separate group. Interestingly, cases with this homozygous mutation disclosed a slowly progressing lower-limb onset phenotype, with mean survival of around 14 years [67]. This distinct profile was marked by a decrease in amino acids in the CSF, as for example arginine, lysine, glutamine, ornithine, serine, and threonine [50]. 

As the first mutated gene described in genetic ALS, mutant *SOD1* is still widely used in genetic ALS models, especially in cellular cultures and rodent models. Thus, metabolomics studies are looking for a better understanding of pathophysiological mechanisms of ALS, which are widely used SOD1 models. Several metabolomics studies performed in cellular models bearing mutations in the *SOD1* gene highlighted a disturbance of energy metabolism. Valbuena and co-workers observed an increased aerobic glycolysis and an amino acid deficit in NSC-34 cells expressing mutant *SOD1-G93A* [51]. The same research team also showed a reduction in glutamate levels in co-cultures of spinal neurons and astrocytes expressing SOD1-G93A, and metabolic alterations related with oxidative stress, involving an important role of neuron–astrocyte relation in ALS pathophysiology [52]. This alteration of glutamate metabolism, which is associated with disturbance of the TCA (tricarboxylic acid) cycle, was also observed in a co-culture model expressing mutant *SOD1-G93A* [68]. To further explore the glutamate hypothesis in ALS, astrocytes and motor neurons bearing *SOD1-G93A* mutation were exposed to glutamate, revealing an alteration of the cellular shuttling of lactate between astrocytes and motor neurons [69]. Metabolic alterations observed in cellular models expressing mutant *SOD1* were confirmed by metabolomics studies performed on animal models such as SOD1 transgenic mice expressing mutants G86R or G93A [53,70,71,72].

Lipids are another class of molecules that are found profoundly altered in ALS patients and can be analyzed trough metabolomics—and more specifically, lipidomics—studies. One study reported a very distinct lipidomics profile in the CSF of ALS patients compared to healthy controls. From the most discriminant lipids, several phosphatidylcholines and sphingomyelins were reported as higher in ALS patients, whereas triglycerides levels were decreased in the CSF of ALS patients. Interestingly, some phosphatidylcholines were also determined as discriminant in the lipidomics profile obtained from the cerebral cortex of the ALS mouse model SOD1-G93A [53,73]. These studies reinforce the notion that even in the presence of pathological mutations, several mechanisms associated with ALS pathology are common between sporadic and familial cases of ALS. Other classes of lipids are also reported to be altered in the plasma of ALS patients, such as triglycerides and fatty acids (reviewed by Gonzalez De Aguilar [74]). Cholesterol seems to be altered in ALS patients, and some studies report hypercholesterolemia at the time of diagnosis [75,76,77,78,79]. It was also shown that the administration of acetyl-L-carnitine, which supports the transport of fatty acids into mitochondria for being used as energy substrate, slowed down the worsening of motor symptoms in ALS patients [80].

### 3.3. TARDBP

Although metabolomics studies were not performed yet in ALS patients bearing mutations in the *TARDBP* gene, in vitro and in vivo studies shed light on the metabolic alterations that could be related with TDP-43 pathology in these models. In a Drosophila model expressing wild type or a mutation associated with ALS (TDP-43-G298S) specifically in motor neurons, metabolomics analysis revealed alterations in the carnitine pathway. They demonstrated an accumulation of carnitine conjugated long-chain fatty acids (as myristoylcarnitine, palmitoylcarnitine, oleoylcarnitine, and linoleoylcarnitine), leading to a decrease in carnitine levels. They also documented a significant reduction in beta-hydroxybutyrate, which is a ketone precursor and a key product of lipid beta-oxidation. These metabolic changes point to defects in the carnitine shuttle, which is required for long-chain fatty acid import into mitochondria and subsequent breakdown by lipid beta-oxidation ultimately leading to ATP production. When they bypassed these deficits by supplementing animals with a mixture of medium-chain fatty acids at varying concentrations (notably, coconut oil), they observed an amelioration in the motor symptoms associated with TDP-43 wild-type and G298S expression [54]. In a cellular model of wild-type TDP-43, metabolomics analysis found alterations in several unsaturated and saturated fatty acids of monounsaturated fatty acids (MUFA), polyunsaturated fatty acids (PUFA), and saturated fatty acids, among others [56]. Another study also performed in the same Drosophila model reported alterations in glucose metabolism, with increase in phosphoenolpyruvate and pyruvate in animals expressing wild-type or mutant TDP-43. Interestingly, when TDP-43 overexpression was knocked down, pyruvate levels return to control levels [55]. Together, these studies demonstrate a clear alteration in the energetic metabolism of ALS patients associated with TDP-43 pathology. 

### 3.4. FUS

A recent metabolomics study performed on motor neurons bearing FUS mutation does not reveal a difference in metabolism due to the presence of FUS mutants [59]. Using iPSC-derived motor neurons from ALS patients carrying *FUS* mutations (R521H or P525L) and their CRISPR/Cas9-corrected counterparts, they highlighted a metabolic switch when cells differentiate to functional motor neurons but did not observe any disturbance of energy metabolism in mutant *FUS* cells compared to controls. It would be interesting to continue this study further by integrating astrocytes or other glial cells.

Ultimately, metabolomics reflects the biochemical activities taking place in an individual in a specific time point, and genetic factors play a key role in the biochemical pathways and ultimately in the production of metabolites [49] (Figure 1). For example, some GWAS applied to metabolic phenotypes generated a wide database of genetically determined metabolites [81,82,83,84]. Very recently, a study used this database in a Mendelian randomization study, combining genomics and metabolomics. They detected 18 metabolites that might have a causal role in the development of ALS and suggest that a dysfunction in the glutathione pathway is involved with ALS pathogenesis. In the same study, they investigated single nucleotide polymorphisms (SNP) in ALS patients and identified several as being genetic variants contributing to ALS pathology. The SNP most associated with ALS was rs1985243, and the authors hypothesize that it might contribute to the development of ALS through affecting the levels of gamma-glutamylphenylalanine [60]. Knowing this, it is possible to follow the alterations in any specific genetic linked case since the non-symptomatic phase and shed a light, at any giving time point, on the biochemical alterations that led the individual to start to experience disease-associated symptoms. This knowledge could help to identify altered metabolites that predict the alterations in molecular processes early in the disease course, and it could further be exploited in non-genetic cases.

One critical feature for the application of metabolomics in combination with genetic analysis in the clinical practice is standardization. Different techniques used in the metabolomics analysis, different methodologies for sample preparation, and even differences in samples storage can induce artificial differences in the metabolome of individuals that are not a real reflect of alterations induced by the disease or the genetic background. This lack of standardization makes it impossible to perform comparisons between studies and different ALS cohorts, and it is probably one of the main problems researchers face when identifying biomarkers for ALS diagnosis or prognosis.

## 4. Combination of Omics and Patient-Derived iPSC: The Future of ALS Research

The development of iPSC reprogrammed from human cells constitutes the ultimate tool when studying diverse pathologies, including neurodegenerative diseases. Nowadays, it is possible to create the “patient in a dish”: iPSC derived from skin fibroblasts or peripheral blood mononuclear cells (PBMC)—or even cells collected from the urine—can be differentiated into virtually all cell types to model disease pathology and study drug response and drug toxicity [85,86,87]. Several research groups started to investigate the pathological mechanisms associated with ALS in iPSC obtained from patients and observe that motor neurons and glial cells differentiated from ALS-iPSC present the same pathological hallmarks as found in post-mortem tissue. Furthermore, they brought to light several disease mechanisms that are common to sporadic and familial or genetic-linked ALS [88,89,90,91].

Although metabolomics analysis in motor neurons or glial cells derived from ALS-iPSC has not been reported yet, several groups investigated the metabolic changes occurring in cells derived from ALS patients when compared to control cells. For example, alterations in the energetic metabolism were also reported in iPSC-derived astrocytes from sporadic and ALS patients bearing the expansions in the *C9ORF72* gene. In these cells, researchers reported defects in adenosine, fructose, and glycogen metabolism [92,93]. The reduction in glycogen metabolism was attributed to decreased mRNA and protein expression of both enzymes glycogen phosphorylase and phosphoglucomutase. They also reported disruptions in the membrane transport of mitochondrial-specific energy substrates [93], which supports the supplementation of ALS patients with drugs that increase the transport of energetic subtracts to the mitochondria, such as the example of acetyl-L-carnitine [80]. Defects in adenosine use as an energy substrate were linked to a reduction in the levels of adenosine deaminase, which is the enzyme that converts adenosine to inosine. Bypassing this defect with inosine treatment restored the metabolism of astrocytes and also inhibited their toxicity when co-cultured with motor neurons. Remarkably, these alterations were also found in some sporadic ALS patients who also presented a decrease in the levels of both mRNA and protein expression of adenosine deaminase [92]. This study also points to another role of inosine treatment: more than increasing the levels of the endogenous antioxidant uric acid, inosine can also be used as an alternative energetic substrate.

Metabolomics analysis performed by different studies showed uric acid to be decreased in several neurodegenerative diseases, such as Parkinson’s disease, Huntington’s disease, and ALS [94,95,96,97,98]. Low uric acid was significantly decreased in the serum of ALS patients compared to control subjects and was inversely associated with an all-cause mortality risk [99]. Interestingly, low urate levels in the plasma were related to a higher risk of developing ALS, years before the onset of symptoms [97]. Preclinical studies showed the protective effect of uric acid [100,101], and uric acid in ALS patients is associated with its important antioxidant properties [102,103,104]. A recent clinical trial evaluated the effects of elevating the levels of urate in the blood of ALS patients by treating them with inosine [105]. Indeed, results recently published demonstrated that the urate levels rose to the targeted levels (7–8 mg/dL) after 6 weeks of treatment, without serious adverse effects observed. Although several biomarkers for oxidative stress and DNA damage were decreased in treated patients, predictions of ALSFRS-R did not change from baseline [105]. Even so, this pilot clinical trial offers hope, as inosine treatment was only performed during 12 weeks, and more prolonged clinical trials with a proper number of patients could reveal important therapeutic effects of inosine. Interestingly, this clinical trial followed the levels of uric acid in treated patients, as both metabolomics markers for treatment strategy and follow up of drug response [106]. Together with data obtained from *C9ORF72*-iPSC [92,93], these data reinforce the need of a better designed clinical trial on the effects of inosine—with, for example, the inclusion of two exclusive arms: one of sporadic ALS patients and another only composed by patients bearing the *C9ORF72* expansions.

Other studies pointed to alterations in the metabolism of cells bearing TDP-43 mutations that are also observed in sporadic cases of ALS presenting cytoplasmic TDP-43 aggregates in motor neurons. iPSC-derived motor neurons from ALS patients bearing a *TARDBP* mutation (G298S) present an increase in the mRNA for two isoforms of the enzyme phosphofructokinase-1 (PFK1), PFKM, and PFKP, which are considered to be rate limiting and control the rate of glycolysis [107]. Remarkably, the same alterations were reported in the Drosophila fly model bearing the same mutation in the *TARDBP* gene (TDP-43-G298S) [55]. Furthermore, PFKM and PFKP transcripts were also increased in spinal cord tissue obtained from sporadic ALS cases with confirmed TDP-43 pathology [55], reinforcing the existence of common pathological mechanisms in sporadic and genetic-linked ALS and that TDP-43 pathology plays a key role in these pathological mechanisms.

Finally, as iPSC derived from ALS patients represent well the pathological characteristics of each patient, these cells could be developed from each patient with the aim to shed light on the pathological alterations taking place in every individual. For example, in the study of Allen et al., not all sporadic ALS patients presented deficits in adenosine deaminase, but these deficits were observed in all C9ORF72 patients. This study strengthens the notion of ALS as a highly heterogeneous disease and the need to develop individual therapeutic strategies, culminating with the establishment of a personalized medicine. Of note, metabolomics techniques are evolving so to be performed in single cells, allowing the study of a single cell metabolome. Some examples of techniques that are sensitive enough to be applied in small volumes and lower concentrations of metabolites are mass spectrometry, matrix-assisted laser desorption ionization (MALDI), and live single-cell mass spectrometry (LSC-MS) [108,109,110,111,112,113]. However, as discussed previously, these methods and techniques need to be standardized to reveal real common alterations among the different ALS subtypes and move the field forward.

## 5. Conclusions

The development of metabolomics analysis in ALS patients with different genetic mutations can help understand the metabolic pathways that are altered in such patients and help understand the pathological mechanisms involved in each case, as ALS is a highly heterogeneous disease. Genetics combined to metabolomics studies can help identify subgroups of patients that could benefit from a particular therapy. Furthermore, metabolomics studies unravel pathological mechanisms that are common to genetic and non-genetic cases and emphasize the impact of TDP-43 pathology in ALS. By unraveling the pathways that are altered in each genetic-linked ALS case, we can envisage a metabolomics-oriented therapy. The combination of metabolomics and iPSC-derived cells from ALS patients will help better understand the pathological mechanisms taking place in each patient and will pave the way to the implementation of a personalized medicine.

## Figures and Tables

**Figure 1 genes-11-00537-f001:**
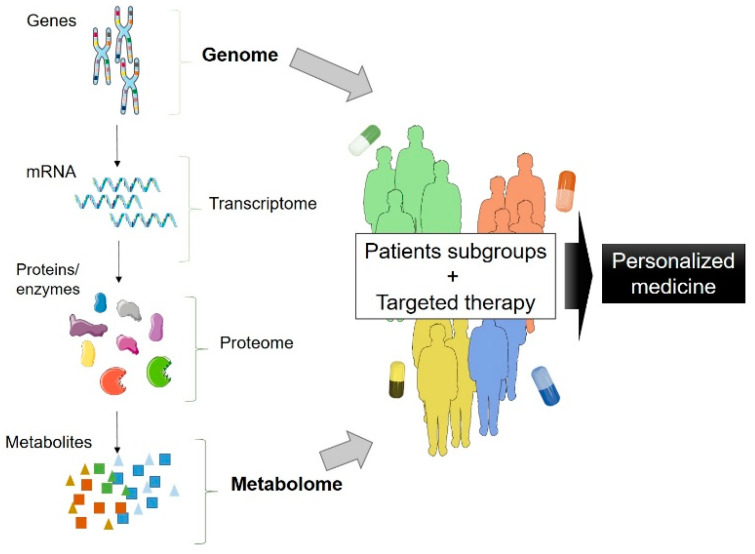
“Omics” approaches analyze the whole set of molecules in an organism, and the metabolome of an individual is ultimately influenced by the genetic background. By combining genotyping and metabolomics approaches, it is possible to identify subgroups of patients and design a targeted therapy for each case. Finally, combination of genetics, metabolomics, and targeted therapy will boost the development of a personalized medicine. The figure was designed using image templates from Servier Medical Art (https://smart.servier.com/image-set-download/).

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
