# Peer review of "Metabolomics: A Tool to Understand the Impact of Genetic Mutations in Amyotrophic Lateral Sclerosis"

_genes, 2020, doi:10.3390/genes11050537_

Round 1

Reviewer 1 Report

In this manuscript entitled "Metabolomics: a tool to understand the impact of genetic mutations in amyotrophic lateral sclerosis", Lanznaster et al review the status of metabolomics in the context of ALS. The authors reviewed the ALS pathology of C9, SOD, Fus and TDP43 and highlighted the scant literature reporting metabolic changes in models of iPSCs from ALS patients. The manuscript is well organized and the content balanced. There are however some minor points that I believe should be reviewed before publication.  Also, although the overall paper is well written, I would encourage the authors to review the text and do some important editing (i.e. "interesting is repeated too many times in the last page). 

The following are the main minor points to be addressed:

Line 101. The authors mention other 19 mutations, are these from the same gene or different genes? What gene/s are these? This would read much better if the nature of these mutations is further explained here.

Line 131. The sentence says "patients with C9 mutations" but it is not explained what mutations it is talking about. It is important to know what this is specifically referring to.It is a similar and recurrent issue that makes the text sound incomplete or superficial.

Line 140. This is exactly the same problem. The sentence talk about SOD mutations without further explaining what mutations it is referring to. And it keeps happening in the whole paragraph while talking about different SOD mutations.

Line 195. The sentence may be more accurate if it says "...in animals expressing both, wild type and mutant TDP43..."

Line 219. iPSC, the abbreviation should be spelled out the first time it is mentioned in the text, rather than at the very end.

Line 266-267. The sentence is not clear. Could the authors clarify whether or not it refers to ALS with mutant TDP-43 pathology versus ALS with wild type TDP43 aggregation?

It would also be interested to know whether the ALS patients enrolled in the clinical trial for Inosine were TDP-43, C9, FUS or SOD1 and what mutations they carried, if this was recorded and the patients were selected for a specific type.

Author Response

In this manuscript entitled "Metabolomics: a tool to understand the impact of genetic mutations in amyotrophic lateral sclerosis", Lanznaster et al review the status of metabolomics in the context of ALS. The authors reviewed the ALS pathology of C9, SOD, Fus and TDP43 and highlighted the scant literature reporting metabolic changes in models of iPSCs from ALS patients. The manuscript is well organized and the content balanced. There are however some minor points that I believe should be reviewed before publication.  Also, although the overall paper is well written, I would encourage the authors to review the text and do some important editing (i.e. "interesting is repeated too many times in the last page). 

Answer: We are thankful for the comments made by this reviewer. We revised the English language and improved the text accordingly.

The following are the main minor points to be addressed:

Line 101. The authors mention other 19 mutations, are these from the same gene or different genes? What gene/s are these? This would read much better if the nature of these mutations is further explained here.

Answer: The sentence was changed to improve meaning (line 104-105).

Line 131. The sentence says "patients with C9 mutations" but it is not explained what mutations it is talking about. It is important to know what this is specifically referring to.It is a similar and recurrent issue that makes the text sound incomplete or superficial.

Answer: we thank the reviewer for bringing this to our attention. We corrected the text to make it clear that the mutation we are discussing regard to the hexanucleotide repeat expansions observed in patients with C9ORF72 mutation (line 77; 145; 157).

Line 140. This is exactly the same problem. The sentence talk about SOD mutations without further explaining what mutations it is referring to. And it keeps happening in the whole paragraph while talking about different SOD mutations.

Answer: we apologize for this recurring fault. We included the description of mutations in SOD1 gene investigated in each study (lines 160-185). We also described the mutations in FUS (line 225) to improve the clarity of our review.

Line 195. The sentence may be more accurate if it says "...in animals expressing both, wild type and mutant TDP43..."

Answer: we thank the reviewer for this suggestion, and we modified the sentence to improve accuracy.

Line 219. iPSC, the abbreviation should be spelled out the first time it is mentioned in the text, rather than at the very end.

Answer: Indeed, iPSC was spelled out the first time it appeared in the text (line 58), but added redundantly at the end, as pointed by the reviewer. We left included the abbreviation in this part of the manuscript (line 253).

Line 266-267. The sentence is not clear. Could the authors clarify whether or not it refers to ALS with mutant TDP-43 pathology versus ALS with wild type TDP43 aggregation?

Answer: The sentence refers to alterations observed in both patients with mutations in the TARDBP gene and sporadic ALS patients that present TDP-43 pathology. We corrected the sentence accordingly (lines 298-300).

It would also be interested to know whether the ALS patients enrolled in the clinical trial for Inosine were TDP-43, C9, FUS or SOD1 and what mutations they carried, if this was recorded and the patients were selected for a specific type.

Answer: We agree with the reviewer. Unfortunately, while in the "Inclusion criteria" depicted the inclusion of both sporadic and familial ALS in the study (https://clinicaltrials.gov/ct2/show/record/NCT02288091), no information on this matter is presented in the study published (Nicholson et al. 2018).

Reviewer 2 Report

The manuscript by Lanznaster et al provides a timely review of the literature describing metabolic changes seen in genetic subtypes of ALS, as well as those seen in cell and animal models of ALS (particularly those developed using ALS-linked mutations). ALS is genetically and phenotypically heterogeneous, with complex etiology. There is hope that metabolomics can differentiate patient subtypes, including those subgroups that are more likely to respond to proposed therapeutic strategies. Changes in metabolic pathways may reflect underlying disease pathways and progression, both of which vary substantially among ALS patient cohorts. It is indeed an attractive idea that metabolic changes may act as a clinical and research tool to better understand the disease. In this context, the study of metabolomics in ALS is in its relative infancy and this review is timely.

Overall, this is a concise yet comprehensive and well written review. There is an insightful Introduction to the known genetic basis of ALS including both familial and sporadic forms of the disease. There is also good balance with review of conflicting reports where evidence for individual metabolic pathways remain contentious. Of course, it remains difficult to ascribe the relative contribution of genetics to an individual’s metabolomic profile, which is evident from this review.

This will be of interest to the field. Nevertheless, the authors should address the following:

The final paragraph of the Introduction is, in part, redundant as it repeats points raised in the previous paragraph.

While it is appropriate to focus on the four common ALS genes, I would have liked to have seen a brief discussion of any data seen in metabolic studies of other ALS subtypes that may support that seen in these four major genes.

It is evident from this review that there is little/no standardised approach to assessing metabolic changes in ALS. Past metabolic studies are diverse and inconsistent across genes. It would be important to add a brief discussion of the need to better facilitate cross-study comparisons. This will be important going forward, particularly for study of ALS iPSC-derived neurons and glia and the promise they hold for better understanding the disease and clinical utility.

It would also be worthwhile discussing how metabolic changes may inform progression of C9orf72-linked ALS to development of cognitive and behavioural deficits leading to clinical of sub-clinical FTD.

Author Response

We are very thankful for the positive comments made by the reviewer.

As pointed out, we modified the last paragraph of the Introduction (page 2), to avoid redundancy.

Unfortunately, there is virtually no studies performing metabolomics analysis in genetic-linked ALS cases induced by others, low frequency variants in genes associated with ALS. One study already discussed in our manuscript, which performed a Mendelian randomization by combining GWAS and metabolomics analysis, reported some single nucleotide polymorphisms (SNP) as genetic variants linked to metabolites alterations and ALS. We included their results regarding SNP in our manuscript (page 6).

As pertinently suggested by the reviewer, we added a discussion about the problems that the lack of standardization in metabolomics approaches across research centers causes in the search for common metabolomics alterations among cases (pages 6-8).

To our knowledge, no study so far reported metabolic changes and metabolomics analysis in ALS/FTLD patients and their relation with motor and cognitive symptoms. However, as lipid metabolism seems to be impaired in models and patients carrying the C9orf72 mutation, we included a discussion about the possible role of lipid metabolism dysfunction and cognitive deficits, as it was reported for other types of diseases characterized by cognitive impairment, as Alzheimer’s disease (page 5).

Reviewer 3 Report

The author, Drs Lanznaster et al., described a review about the metabolomics as a tool to understand the impact of genetic mutations in amyotrophic lateral sclerosis.

Major points

The manuscript lacks attractive figures and relevant tables. The Author should add impressive images and informative tables for better understanding. Moreover, the text should include a brief section on autosomal recessive forms of ALS and their implications for metabolomics. ALS5/SPG11 should be discussed (Montecchiani et al, BRAIN 2016). Several references are too old and should be updated. Finally, the quality of written English needs some language corrections.

Author Response

We thank the reviewer for her/his suggestions. We added a figure (page 4) and a table (page 3) to our manuscript, as suggested.

Unfortunately, there is virtually no studies performing metabolomics analysis in genetic-linked ALS cases induced by others, low frequency variants in genes associated with ALS. Regarding the study of Montecchiani et al, they presented data from individuals with Charcot-Marie-Tooth disease with mutations in the ALS5 gene, which is also associate with juvenile forms of ALS. Although they performed blood tests and CSF examination, they only mentioned in the study that all results were in the “normal range”. However, no metabolic or metabolomics investigation was performed in this study. Nevertheless, one study included in our manuscript, which performed a Mendelian randomization by combining GWAS and metabolomics analysis, reported some single nucleotide polymorphisms (SNP) as genetic variants linked to metabolites alterations and ALS. Therefore, we also included these results in our manuscript.

In the regard of the references, the only references that are indeed old regard the studies that described for the first time the alterations or the role of the proteins discussed in the text (an account for around 10% of all the cited studies). We also improved the English language accordingly.

Round 2

Reviewer 3 Report

The authors did not fulfill the criticisms. Thus, the manuscript is still immature for publication